# Transcription Factor Prospero Homeobox 1 (PROX1) as a Potential Angiogenic Regulator of Follicular Thyroid Cancer Dissemination

**DOI:** 10.3390/ijms20225619

**Published:** 2019-11-10

**Authors:** Magdalena Rudzińska, Michał Mikula, Katarzyna D. Arczewska, Ewa Gajda, Stanisława Sabalińska, Tomasz Stępień, Jerzy Ostrowski, Barbara Czarnocka

**Affiliations:** 1Centre of Postgraduate Medical Education, Department of Biochemistry and Molecular Biology, 01-813 Warsaw, Poland; magdda.rudzinska@gmail.com (M.R.); katarzyna.arczewska@cmkp.edu.pl (K.D.A.); ewa.gajda@cmkp.edu.pl (E.G.); 2Institute of Molecular Medicine, Sechenov First Moscow State Medical University, 119991 Moscow, Russia; 3Maria Skłodowska-Curie Memorial Cancer Center and Institute of Oncology, Department of Genetics, 02-781 Warsaw, Poland; mikula.michal@gmail.com (M.M.); jostrow@warman.com.pl (J.O.); 4Nalecz Institute of Biocybernetics and Biomedical Engineering, Polish Academy of Sciences, 02-109 Warsaw, Poland; ssabalinska@ibib.waw.pl; 5Clinic of Endocrinological and General Surgery, Medical University of Łódź, 91-513 Łódź, Poland; tomsamaz@wp.pl; 6Centre of Postgraduate Medical Education, Department of Gastroenterology, Hepatology and Clinical Oncology, 02-781 Warsaw, Poland

**Keywords:** *PROX1*, angiogenesis, follicular thyroid cancer

## Abstract

It is well known that Prospero homeobox 1 (PROX1) is a crucial regulator of lymphangiogenesis, that reprograms blood endothelial cells to lymphatic phenotype. However, the role of PROX1 in tumor progression, especially in angiogenesis remains controversial. Herein, we studied the role of PROX1 in angiogenesis in cell lines derived from follicular thyroid cancer (FTC: FTC-133) and squamous cell carcinoma of the thyroid gland (SCT: CGTH-W-1) upon *PROX1* knockdown. The genes involved in angiogenesis were selected by RNA-seq, and the impact of PROX1 on vascularization potential was investigated using human umbilical vein endothelial cells (HUVECs) cultured in conditioned medium collected from FTC- or SCT-derived cancer cell lines after PROX1 silencing. The angiogenic phenotype was examined in connection with the analysis of focal adhesion and correlated with fibroblast growth factor 2 (FGF2) levels. Additionally, the expression of selected genes involved in angiogenesis was detected in human FTC tissues. As a result, we demonstrated that *PROX1* knockdown resulted in upregulation of factors associated with vascularization, such as metalloproteinases (MMP1 and 3), FGF2, vascular endothelial growth factors C (VEGFC), BAI1 associated protein 2 (BAIAP2), nudix hydrolase 6 (NUDT6), angiopoietin 1 (ANGPT1), and vascular endothelial growth factor receptor 2 (KDR). The observed molecular changes resulted in the enhanced formation of capillary-like structures by HUVECs and upregulated focal adhesion in FTC-133 and CGTH-W-1 cells. The signature of selected angiogenic genes’ expression in a series of FTC specimens varied depending on the case. Interestingly, *PROX1* and *FGF2* showed opposing expression levels in FTC tissues and seven thyroid tumor-derived cell lines. In summary, our data revealed that PROX1 is involved in the spreading of thyroid cancer cells by regulation of angiogenesis.

## 1. Introduction

Thyroid cancer malignancies are divided into three major types: (1) differentiated thyroid carcinoma (DTC) arising from follicular cells of the thyroid, (2) anaplastic thyroid carcinoma, and (3) medullary thyroid carcinoma arising from parafollicular C cells. DTC is the most common thyroid malignancy and accounts for 90%–95% of all thyroid cancer cases [1]. Two predominant histological groups of DTC are papillary (PTC) and follicular (FTC) thyroid carcinoma, accounting for ~80% and ~15% of DTCs, respectively [2]. On the other hand, squamous cell carcinoma of the thyroid (SCT), constituting less than 1% of all thyroid malignancies, is an aggressive neoplasm thought to arise as a primary tumor or as a component of an anaplastic or undifferentiated carcinoma and gives the distant metastasis [3]. The thyroid gland is a highly vascularized organ with increased vascularity observed in thyroid diseases, including tumors [2]. Depending on the type, DTCs spread through different pathways and PTCs have a propensity to disseminate via lymphatic vessels to the neck regional lymph nodes, whereas FTCs tend to metastasize to remote organs by the hematogenous route [4]. Metastases from primary SCT are common and mostly its vascular invasion occurs in the lungs, bones, liver, kidney, and heart [3].

Angio- and lymphangiogenesis are closely related processes, with hematopoietic and lymphatic factors expressed at different levels in both lymphatic endothelial cells (LECs), and blood vessels endothelial cells (BECs) [5]. The formation of new vessels is strongly activated in cancer development. Following this observation, recent studies underlined the significant role of angiogenesis and lymphangiogenesis in cancer progression and indicated the therapeutic potential for the prevention of these processes [5].

Transcription factor Prospero homeobox 1 (PROX1) is a mammalian homolog of *Drosophila homeobox* protein Prospero [6] and is vital for embryonic development of organs, e.g., the central nervous system, heart, lens, retina, liver, pancreas, and lymphatic vascular system [7,8,9,10,11,12,13]. As a marker for mammalian lymphatic endothelial cells, PROX1 is expressed in a subpopulation of endothelial cells that give rise to the lymphatic system [13]. Additionally, PROX1 is described as a regulator of vascular endothelial growth factor VEGF receptor-3 (VEGFR-3) and lymphatic vessels endothelial hyaluronan (LYVE-1), which are strongly involved in the lymph- and angiogenesis [14]. 

PROX1 is significantly engaged in tumorigenesis and plays various tissue-dependent functional roles in cancer dissemination. It acts as a tumor suppressor in hematologic malignancies, breast cancer, esophageal cancer, pancreatic cancer, and carcinomas of the biliary system [15,16,17,18,19], to name a few. However, other reports have demonstrated that the upregulation of PROX1 is a predictor of poor outcomes in colon cancer, glioblastoma, and vascular endothelial tumors [20,21,22]. A recent study showed that PROX1 might affect the malignant phenotype of colorectal tumor cells by regulating angiogenesis [23]. 

Our previously published data showed that transcription factor PROX1 is strongly expressed in FTC-133 and CGTH-W-1 compared to PTC-derived cell lines, which further suggests a possible relationship between PROX1 expression and potential of more aggressive thyroid cancer metastasis via the blood system [24]. 

In the present study, we aimed to evaluate the potential involvement of PROX1 in the regulation of thyroid cancer angiogenesis. Thus, by comparing transcriptomic profiles of FTC and SCT-derived cells after PROX1 silencing and cells treated with control siRNA, we observed the activation of many angiogenic factors, that induce intensified endothelial tube formation. Furthermore, we correlated the observed phenotype with enhanced focal adhesion, which is an integral part of angiogenesis [25].

Finally, we demonstrated that PROX1 and other vascular factors, such as VEGFC (vascular endothelial growth factor C), BAIAP2 (BAI1 associated protein 2), FGF2 (fibroblast growth factor 2), and PLAT (plasminogen activator) are differently expressed in FTC human tissues compared to non-tumor tissues. However, in all tested thyroid cancer cell lines and tissues of different origins, we observed the inverse PROX1:FGF2 relation. Interestingly, the treatment of CGTH-W-1 with FGF2 resulted in the higher expression of PROX1, which indicates mutual regulation of PROX1 and FGF2 signaling generating a regulatory loop in thyroid cancer cells.

Taken together, our study thereby describes a new molecular mechanism, which can be fundamental in metastasis of aggressive thyroid cancers.

## 2. Results

CGTH-W-1 and FTC-133 cells were transfected with siRNAs targeting *PROX1* (si*PROX1*) and with a control universal negative siRNA (siNEG). The down-regulation of PROX1 was evaluated using RT-qPCR, Western blotting, and immunofluorescence methods. Only ~2% of the initial *PROX1* transcript level was detected in CGTH-W-1 and FTC-133 cells 48 h after transfection (Figure 1a,b). Western blotting and immunofluorescence assays demonstrated the knockdown of PROX1 to almost undetectable levels in both cell lines analyzed. These observations are in agreement with our previously published data [24,26], and here we again confirm the effect of silencing of PROX1 protein using Western blotting for si*PROX1* purchased from both sources (i.e., Sigma Aldrich and Santa Cruz; Appendix A).

### 2.1. PROX1 Silencing in CGTH-W-1 Cells Has a Significant Impact on the In Vitro Angiogenesis

As we reported recently, *PROX1* depletion in CGTH-W-1 cells results in differential expression of 1182 genes in comparison to control cells treated with siNEG in RNAseq analysis [26]. Of these, 55 genes (Appendix A) identified using two silencers (si*PROX1* SA and SC) were present on the list of 478 genes related to angiogenesis described by Chu LH. et al. (2012) [27]. Using RT-qPCR we confirmed the increased expression of several genes: *MMP1*, *MMP3* (matrix metallopeptidase 1 and 3), FGF2, *ANGPT1* (angiopoietin 1), *BAIAP2*, *KDR* (kinase insert domain receptor), *VEGFC*, NUDT6 (nudix hydrolase 6), and decreased expression of *ADAMTS9* (ADAM metallopeptidase with thrombospondin type 9), *MDK* (midkine), *VEGFA* (vascular endothelial growth factor A), *PLAT*, and *TIMP3* (TIMP metallopeptidase inhibitor 3) in CGTH-W-1 cells upon *PROX1* depletion (Figure 1a). The upregulation of *MMP1, FGF2, TIMP3, KDR, ANGPT1, MMP3, NUDT6, BAIAP2, VEGFC* was also confirmed after silencing of *PROX1* in FTC-133 cells (Figure 1b), whereas the expression of other analyzed genes was not significantly changed. The only gene whose expression after PROX1 silencing significantly differed between CGTH-W-1 and FTC-133 cells was *TIMP3*. This difference was probably due to the difference in the variability of regulation and basic expression of MMPs and TIMPs in thyroid tumor cells.

In cells with the increased level of FGF2 transcript, Western blotting analysis showed a significantly higher level of FGF2 protein in CGTH-W-1 and FTC-133 cells after the PROX1 silencing in comparison to control treated cells. The significantly increased level of VEGFC protein in cell lysates and the medium was observed only for CGTH-W-1 (Figure 1c).

To investigate the pro-angiogenic effect of factors secreted to the medium upon PROX1 silencing, we performed the tube formation assay with human umbilical endothelial cells (HUVECs) using si*PROX1* (SA) (Figure 2a,b) or si*PROX1* (SC) (Appendix A). HUVECs were incubated on Matrigel-coated plates in conditioned medium collected from cancer cells with PROX1-silenced or control cells treated with siNEG. In applied experimental setup, HUVECs revealed the behavior of endothelial cells influenced by regulators secreted from tumoral cells. In particular, tubule formation by HUVECs was stimulated by conditioned medium collected from both CGTH-W-1 and FTC-133 cells after *PROX1* silencing in comparison to conditioned medium collected from control cells treated with siNEG. The above phenotype paralleled higher expression levels of pro-angiogenic genes and proteins observed upon *PROX1* knockdown. Further supporting this phenomenon, the increased number of branching (the nodes connected to tree different line segments, which indicates the new vessels sprout), meshes/loops, and junctions between the endothelial cells was observed after PROX1 silencing (Figure 2c,d) indicating that the absence of PROX1 enhances the new blood vessel formation.

In the previous study, we observed significant changes in a number of molecules regulating the focal adhesion (e.g., caveolins, FAK kinase, integrins, collagens, and chemokines), which is a feature tightly linked to vascularization. Here, we analyzed focal adhesions (FAs) of CGTH-W-1 and FTC-133 thyroid cancer cells after PROX1 silencing in comparison to control cells treated with siNEG. We observed a significant difference in the number of FAs in both cases, CGTH-W-1 (*p* < 0.01) and FTC-133 (*p* < 0.01). The difference in the size of FAs was comparable in treated and control cells and was estimated as ~1 µm and ~4 µm for CGTH-W-1 and FTC-133 cells, respectively (Figure 3).

### 2.2. Genes Involved in Angiogenesis (FGF2, VEGFC, PLAT, BAIAP2) are Variably Expressed in Follicular Thyroid Cancer Tissues. Higher Expression of PROX1 Corresponds to a Lower Stage of Thyroid Cancer and is Negatively Correlated to FGF2 Expression.

To investigate the angiogenic factors in FTCs, we selected a few genes which are actively implicated in vascularization: *FGF2, VEGFC, BAIAP2, PLAT* and we defined their expression in human FTC tissues (T) and paired healthy tissues (NT) by RT-qPCR (Appendix A). The analysis revealed that their expression levels in the FTCs group varied from one case to another and the same fluctuation was previously observed for PROX1 [26]. Data showed increased expression of FGF2, VEGFC, and BAIAP2 in tumoral tissue in six, five, and four FTC samples, respectively, whereas the same genes were overexpressed in five, six, and six NT tissues, respectively. PLAT revealed higher expression in three tumor cases, six healthy tissues, and two cases had a similar PLAT expression level. These data show a considerable variation of angiogenic factors expression in patient’s tissues, but it should be underlined that the thyroid gland (healthy tissue), as well as a tumor region, are highly vascularized, which may affect the expression of pro-angiogenic genes. Therefore, the T-to-NT ratio was calculated and as a result we observed higher expression of PROX1 in three tumor cases (case number: 42, 149 and 157), whereas FGF2 expression revealed opposite pattern. In eight cases that showed PROX1 downregulation in the tumor tissue, the FGF2 expression fold change was higher than PROX1 (Figure 4a). 

Then, using a number of cell lines from various thyroid cancer subtypes, we checked whether the negative correlation of PROX1 and FGF2 expression is a characteristic feature of FTCs or whether it occurs also in other thyroid cancers, such as PTCs and anaplastic cancer. We analyzed the expression of both genes in the panel of cell lines from SCT (CGTH-W-1), FTC (FTC-133, FTC-236, FTC-238), PTC (TPC1, BcPAP), and anaplastic carcinoma (8505C) in comparison to the cell line from the normal thyroid gland (NTHY-ori 3-1) (Figure 4b). Interestingly, a similar opposing pattern of PROX1:FGF2 expression was observed in all analyzed cancer cell lines, but not cells derived from normal thyroid. 

FTC is a relatively rare form of thyroid cancer, so we analyzed results of PROX1 and FGF2 expression in the PTC subtype available through the GEPIA database (Gene Expression Profiling Interactive Analysis). Using this tool, we observed the negative correlation between PROX1 and FGF2 expression in PTC (Figure 4c), but this was not observed for normal thyroid tissues (data not shown), which is in line with our data from NTHY-ori 3-1 cells.

Subsequently, we compared the expression level of PROX1 between the tumor stages and its influence on Kaplan–Meier survival curves. The analysis revealed that PROX1 is upregulated in I and II PTC stages, and decreased in III and IV stages, whereas the expression of FGF2 shows the opposite direction of expression pattern (Figure 4d). Moreover, patients with higher PROX1 expression had reduced survival probability (Appendix A). This observation may suggest that PROX1 can support cancer development, but in the more aggressive stage of the tumor, PROX1 is downregulated at the cost of higher expression of FGF2. This molecular connection can be case-dependent and likely considered as a predictor marker for tumor stage and disease prognosis.

### 2.3. Treatment of CGTH-W-1 Cells with Pro-Angiogenic FGF2 Upregulates PROX1 Expression

To further validate angiogenesis and PROX1:FGF2 association, we incubated CGTH-W-1 cells with FGF2, which is a strongly mitogenic, and pro-angiogenic factor. Cells were incubated 24 h with FGF2 (Figure 5). The results revealed that FGF2 treatment enhances the PROX1 mRNA (4-fold change) and protein expression level in CGTH-W-1 cells in comparison to control cells treated with 0.01% DMSO. Interestingly, the expression levels of PLAT, ANGPT1, BAIAP2, NUDT6, PTK2, and VEGFC mRNAs were regulated in the opposite direction in comparison to changes induced by PROX1 silencing. In particular, PLAT, which was downregulated after PROX1 knockdown, revealed higher expression after FGF-2 treatment, whereas ANGPT1, BAIAP2, NUDT6, PTK2, and VEGFC were decreased upon FGF2 treatment. As previously observed in rat lenses treated with FGF transcriptional responses induced by FGF2, including PROX1 upregulation, may be regulated by the MAPK kinase pathway. This effect powerfully highlights the close link and mutual regulation between PROX1 and FGF2 in thyroid cancer-derived cells.

## 3. Materials and Methods

Functional studies were performed on the cell lines CGTH-W-1 (thyroid gland squamous cell carcinoma (derivative of SW-579); originally thought to be established from FTC metastasizing to the sternum), BcPAP (PTC) and 8505C (anaplastic thyroid carcinoma) purchased from the German Collection of Microorganisms and Cell Cultures (DSMZ, Braunschweig, Germany, ref. ACC-360, ACC 273 and ACC 219, respectively), FTC-133 (FTC; metastasis to lymph node), FTC-236 (FTC; metastasis to neck lymph node), FTC-238 (FTC; metastasis to lung), as well as Nthy-ori 3-1 cells (thyroid follicular epithelial cells) obtained from the European Collection of Cell Cultures (ECACC, UK, ref. 94060901, 06030202, 94060902 and 90011609, respectively). TPC1 cells were kindly provided by Dr. M. Santoro (The University of Naples Federico II, Italy). The cells were cultivated in complete Roswell Park Memorial Institute medium 1640 (RPMI-1640) or in Dulbecco’s modified Eagles’s medium (DMEM):F-12 (1:1), supplemented with 10% fetal bovine serum (Gibco, Grand Island, NY, USA). For in vitro angiogenesis assay, we used human umbilical vein endothelial cells (HUVECs), kindly provided by Dr. G. Hoser (Laboratory of Flow Cytometry, Centre of Postgraduate Medical Education Warsaw, Poland). The HUVEC cells were cultivated in Endothelial cell growth medium MV2 (PromoCell, Heidelberg, Germany), containing 5% fetal calf serum, 5 ng/mL epidermal growth factor, 10 ng/mL basic fibroblast growth factor, 20 ng/mL insulin-like growth factor, 0.5 ng/mL vascular endothelial growth factor, 1 µg/mL ascorbic acid, 0.2 µg/mL hydrocortisone. All cells were incubated at 37 °C in a humidified 5% CO_2_ atmosphere.

The series of 11 fresh samples of human follicular thyroid carcinomas (T) and 11 adjacent normal thyroid tissue from the contralateral lobe (NT) were obtained from patients with sporadic FTC undergoing surgical resection at the Department of General and Endocrinological Surgery, Copernicus Memorial Hospital (Łódź, Poland). The samples were snap-frozen in liquid nitrogen and stored at −75 °C. The pairs of tissue numbers were assigned internally, and they were 77, 74, 34, 38, 27, 42, 117, 3, 122, 149, 157 T and NT. Informed consent was obtained from all patients, and the work was approved by ethical committees at the Center of Postgraduate Medical Education, and at the Copernicus Memorial Hospital (RNN/135/KE; 15/7/2014). 

### 3.1. RNA Extraction and cDNA Synthesis

Total RNA was extracted from human follicular thyroid specimens and cells using Universal Purification Kit (EURx, Gdansk Poland), followed by on-column DNAse (A&A Biotechnology, Gdynia, Poland) digestion. The quality of RNA samples was determined using Synergy 2 Multi-Mode Reader (BioTek, Vinooski, VT, USA). Complementary DNA (cDNA) was transcribed from mRNA using High-Capacity cDNA Reverse Transcription Kit (Life Technologies, Applied Biosystems, Foster City, CA, USA), according to the manufacturer′s protocols. For RT reaction 1 µg of total RNA was used with optical density OD260/OD280 1.7–2.0.

### 3.2. Real-Time Quantitative PCR

Expression of the human genes was quantified by RT-qPCR, where the cDNAs served as a template using Maxima Fluorescein RT-qPCR Master Mix (ThermoFisher Scientific Waltham, MA, USA), containing the double-stranded DNA-specific dye SYBR Green I, and specific oligonucleotide primers (primer sequences for RT-qPCR are presented in Appendix A). PCR reactions were performed in triplicates with the following conditions: 95 °C/30 s, 40 cycles of 95 °C/5 s, 58 °C/15 s and 72 °C/10 s in iQ5 Real-Time PCR Detection System (Bio-Rad, Hercules, CA, USA). The Ct values estimated for analyzed genes were normalized against corresponding Ct values of *β-ACTIN*.

### 3.3. Western Blotting

Protein lysates collected from cells were prepared in NP-40 lysis buffer (150 mM sodium chloride; 1.0% NP-40; 0.5% sodium deoxycholate; 0.1% SDS; 50 mM Tris, pH 8.0) supplemented with 1× protease and phosphatase inhibitor cocktails (Roche, Basel, Switzerland). Next, the protein concentrations in lysates were determined using the Bradford assay. Supernatants collected from the same number of cells were supplemented with 1× protease and phosphatase inhibitor cocktails, and media were further processed in parallel with protein lysate. (Sigma-Aldrich, St. Louis, MO, USA). Lysates (30 μg of protein and 25 μL of medium per well) were resolved in 10% SDS–PAGE gels and subsequently transferred to nitrocellulose membranes (Bio-Rad, Hercules, California, USA). The blots were probed with the goat anti-human PROX1 (1:2000; cat no. AF752 R&D Systems, Minneapolis, MN, USA), anti-human VEGFC (1:2000; cat no. AF2727 R&D Systems, Minneapolis, MN, USA), and anti-FGF2 (1:1000; cat no. SAB2100814 Sigma Aldrich, Saint Louis, MO, USA) primary antibodies, followed by HRP Rabbit Anti-Goat IgG (1:20000; cat no. 305-035-046, Jackson ImmunoResearch, West Grove, PA, USA) or HRP-conjugated affinity-purified Goat Anti-Rabbit IgG (1:5000; cat no. P0448 DAKO, Carpinteria, CA, USA) secondary antibodies. Signals from reactive bands were visualized by enhanced chemiluminescence detection (SuperSignal^®^ West Dura, Pierce Chemical, ThermoFisher Scientific, Rockford, IL, USA). As a loading control, the membranes were incubated with primary monoclonal anti-β-actin antibody (1:5000; cat no. A2228 Sigma-Aldrich, USA) followed by goat anti-mouse IgG (1:20,000; cat no. 115-035-003 Jackson ImmunoResearch Laboratories) in an identical manner.

### 3.4. Transient Transfection of Small Interfering RNA

Cells were transfected with siRNA (final concentration 30 nM) targeting human *PROX1* (MISSION esiRNA human PROX1, Sigma-Aldrich, USA, termed further si*PROX1* (SA) or with *PROX1* siRNA (h), sc-106451, Santa Cruz Biotechnology, California, USA, termed below si*PROX1*(SC)) using Lipofectamine 2000 (Life Technologies, Carlsbad, CA, USA) in Opti-MEM medium (ThermoFisher Scientific, Waltham, MA, USA), according to manufacturer′s recommendations. Scramble siRNA was used as the negative control (siNEG; MISSION siRNA, SIC-001, Sigma-Aldrich, USA). The experiments were conducted in triplicates and at least three times. Before further processing cells were silenced for 48 h unless otherwise indicated. Each time *PROX1* knockdown was verified by RT-qPCR.

### 3.5. Tube Formation Assay

For CGTH-W-1 and FTC-133 cells, endothelial tube formation assays were performed in 96-well Matrigel-coated tissue culture plates (Becton-Dickinson, Franklin Lakes, NJ, USA) (1 mg/mL). Matrigel pre-coated plates were incubated at 37 °C for 30 min to allow polymerization. Subsequently, 2 × 10^4^ human umbilical vein endothelial cells (HUVEC) were suspended in conditioned medium collected from CGTH-W-1 or FTC-133 cells after 48 h treatment with siPROX1 or control siNEG and seeded on Matrigel layer. After 5 h incubation, the tube formation was imaged under inverted microscope AxioObserver D1 (Carl Zeiss, Oberkochen, Germany), magnification 100× and 200×.

### 3.6. Angiogenesis Analysis

The angiogenic parameters were examined using Angiogenesis Analyzer macro connected Image J software (Gilles Carpentier, http://image.bio.methods.free.fr/ImageJ/?Angiogenesis-Analyzer-for-ImageJ).

### 3.7. Culture with FGF2

First, 1 × 10^4^ CGTH-W-1 cells were cultured for 24 h with 1 µg/mL FGF2 (Cell Signaling, Danvers, MA, USA) or in control conditions with 0.01% DMSO alone (Sigma-Aldrich, USA). After incubation, total RNA and protein samples were isolated. Next, PROX1 mRNA and protein expression levels were analyzed using RT-qPCR and WB techniques, respectively.

### 3.8. Protocols of Focal Adhesion Measurement

The measurement was based on the processing of immunofluorescence images performed on PROX1-silenced CGTH-W-1 and FTC-133 cells with a primary antibody against FAK kinase phosphorylated at Y397 (an active form of FAK). The detailed protocol of immunostaining was described previously [24]. All images were processed using the ImageJ software and plugins (CLAHE and Log3D). Measurement included seven steps: (1) subtracting background with sliding paraboloid option with rolling ball radius = 50 pixels, (2) enhancement of local contrast of the image by running clahe (values: block size = 19, histogram bins = 256, maximum slope = 6, no mask and fast), (3) minimalization of background using mathematical exponential (exp) option, (4) adjusting the brightness and contrast automatically, (5) running log3d (Laplacian of Gaussian or Mexican Hat) filter, where sigma X and Y = 5, (6) automatic threshold adjustment, (7) analyzing particles (with parameters size = 50-infinity and circularity = 0.00–0.99).

### 3.9. Statistical Analysis

All experiments were repeated at least three times. Data are reported as means ±SD. Data were analyzed using the nonparametric Mann–Whitney U test (GraphPad, Prism 6.00 for Windows, Graf Pad software, San Diego, CA, USA). A *p* value of <0.05 was considered statistically significant. 

## 4. Discussion

The close association of lymphatic and blood vessel development suggests that some factors may control both processes: angio- and lymphangiogenesis. 

Clinicopathological data indicate that DTCs spread by different pathways, and it is assumed that FTC cells spread mainly via the vascular system, the same as SCT. However, in a previous study, we detected the higher expression level of the lymphatic marker-PROX1 in CGTH-W-1 and FTC-133 cell lines compared with cells derived from PTCs (BcPAP, TPC1) and immortalized normal thyroid cells (NTHY-ori 3-1) [24]. In the present study, we consider that PROX1 can be involved in distant metastasis and regulation of the vascular system. Moreover, we observed that this process is closely related with FGF2. 

It is well known that PROX1 is a key factor in maintaining the normal phenotype of lymphatic endothelial cells. The absence of PROX1 activates the expression of pro-angiogenic markers in LEC cells whereas PROX1 overexpression is sufficient to convert the phenotype of BEC cells into lymphatic phenotype [13,29]. Consequently, the lack of PROX1 leads to molecular rearrangement in endothelial cells and enhancement of BEC phenotype [30]. The experiments reveal that the deletion of *PROX1* gene in LECs leads to a decrease of expression of lymphatic markers, including podoplanin, while the level of vascular markers, such as CD34 and endoglin is significantly increased. Moreover, the PROX1 knock-down in mice lens fibers induces the upregulation of genes involved in blood vessel development and focal adhesion signaling pathways, as well as downregulation of genes involved in binding of cytoskeleton proteins [31]. The opposite effect was observed in mouse immature B-cells, where PROX1 overexpression induced a decrease of several FGF signaling pathway members, including downregulation of FGF2 expression level [32]. However, the mutual regulation of PROX1 and FGF2 was never investigated in human cancer.

Results presented in the current study indicate that changes in PROX1 expression in CGTH-W-1 and FTC-133 cells may affect genes involved in angiogenesis, which were correlated in the “angiome interaction network” described by Chu et al., where authors created the list of the genes involved in vascularization [27]. The silencing of PROX1 leads to up- or downregulation of 55 angiogenic genes in CGTH-W-1 selected using RNAseq. The expression of several transcripts, such as *MMP1*, *FGF2*, *MMP3*, *NUDT6*, *BAIAP2*, *VEGFC*, *ANGPT1,* and *KDR* was significantly increased, whereas *VEGFA*, *ADAMTS9*, *MDK*, TIMP3, and *PLAT* were significantly suppressed. qRT-PCR analysis confirmed overexpression of the subset of the genes, including *MMP1, FGF2, TIMP3, KDR, ANGPT1, MMP3, NUDT6, BAIAP2,* and *VEGFC* in FTC-133 cells. The difference in TIMP3 expression after PROX1 silencing in CGTH-W-1 and FTC-133 is probably a result of opposing regulation and variable baseline expression of MMPs and their inhibitors in cell lines derived from thyroid carcinomas [33].

Overexpressed proteolytic enzymes MMP1 and MMP3 romote tumor growth and vascularization by enhancing extracellular matrix degradation in tumor cell microenvironments [34,35,36]. The important trigger of proteinase cascade that results in the generation of high local concentrations of plasmin and active MMPs is the plasminogen activator *PLAT* [37]. MMPs are subject to negative regulation leading to inactivation by specific inhibitors—TIMPs. TIMP3 may block the binding of VEGF to VEGF receptor-2 and inhibit downstream signaling and angiogenesis [38]. Next, decreased extracellular protease ADAMTS9 plays anti-angiogenic and tumor suppressive functions, e.g., in esophageal squamous cell carcinoma [39]. 

Inhibition of glycoprotein-angiopoetin 1 (ANGPT1) expression leads to reduced vascular system integrity [40]. Vascular endothelial growth factors with their receptors (VEGFC/A, VEGFR-1/3) have been characterized as essential lymphangiogenic factors, but research also showed their important role in angiogenesis and tumor progression via blood vessels [41,42,43].

FGF2 is a member of the fibroblast growth factor family, which has mitogenic activity and activates angiogenesis in vitro and in vivo [44]. Antisense of the *FGF2*-*NUDT6* gene overlaps and lies on the opposite strand of *FGF2* mRNA and thus was proposed to regulate FGF2 expression [45]. 

Brain-specific angiogenesis inhibitor (BAI1)-binding protein-BAIAP2 is a member of the adhesion GPCR subfamily, highly expressed in the normal brain tissue and epigenetically silenced in glioblastoma [46]. This protein is involved in the neurogenesis signaling pathway, which was also significantly regulated after PROX1 knockdown in CGTH-W-1 cells. Moreover, the importance of neurogenic molecules in thyroid carcinoma development was previously observed with a neuron- glia-related cell-adhesion molecule (NrCAM) [47,48], which can suggest that BAIAP2 may act in a similar manner 

Next, midkine (MDK) is a secreted growth factor that binds heparin and responds to retinoic acid, as well as promotes cell growth, migration, and angiogenesis, in particular during tumorigenesis [49].

Additionally, in a previous study performed on FTC-133 and CGTH-W-1 cells [24,26] we have shown that PROX1 can control the expression of adhesive molecules, including protein tyrosine kinase 2 (PTK2), caveolins (CAV-1,-2), integrins (2 and 11), collagens (6, 16 and 18) that are also involved in the regulation of angiogenesis [25,50,51,52]. The link between focal adhesion and angiogenesis is well established [53]. Herein we confirmed the enhanced focal adhesion after silencing of PROX1 using the measurement of FAs and as a marker of FAs, we used the active form of FAK kinase that has also been implicated as an essential modulator of angiogenesis [54]. 

In order to elucidate the impact of the observed molecular changes, we performed phenotypical analysis. We observed that HUVEC cultured in conditioned medium collected from CGTH-W-1-PROX1 and FTC-133-PROX1 cells had a stronger influence on tube formation than medium from control cells. These results suggest that PROX1 suppression stimulates the secretion of a number of proangiogenic substances, including, for example, VEGFC and/or FGF2, and consequently promotes angiogenesis in the tumor microenvironment [55]. This phenotype has already been observed in knockout mice models by Wigle et al. (2002) [14] and Johnson NC et al. (2008) [30], where PROX1 silencing increased the secretion of proangiogenic factors.

In experiments performed by Korah R and colleagues, the higher expression of FGF2 induced enhanced focal adhesion, which was accompanied by lower motility of breast cancer cells [56]. Furthermore, it was shown that FGF2 could stimulate ERK activity and FAK phosphorylation [57]. Here we observed that PROX1 expression is regulated in the opposite direction than FGF2, as well as that PROX1 silencing increases focal adhesion. Based on these observations, we can hypothesize that changes in focal adhesion are triggered by increased FGF-2 expression. Interestingly, this accumulation of FGF-2 may be relevant to cancer cells in metastatic microenvironments, where it may contribute to a dormant state of tumor cells [58,59].

On the other hand, our observations may result from a compensatory effect, where cancer cells after losing expression of an important vascular factor respond by upregulating expression of other angiogenic factors [60]. 

Then, we investigated the expression of *FGF2*, *VEGFC*, *BAIAP2,* and *PLAT* in human FTCs and normal tissues. Expression of individual factors was highly variable among patients and difficult to normalize due to high vascularization of tumoral thyroid tissue, as well as the normal thyroid gland. The molecular pattern of angiogenic factors suggested that angiogenesis is strongly committed in the dissemination of FTC and SCT to distant organs. 

Interestingly, we observed an interdependence of PROX1 and FGF2 expressions. The follicular carcinomas, the analyzed cell line derived from FTC cancer, as well as the cell line derived from SCT, papillary, and anaplastic carcinomas showed a similar correlation. This association was not observed in normal thyroid tissues or in the normal thyroid cell line. Moreover, the analysis performed using the GEPIA database also revealed the same relation between PROX1 and FGF2 expression in a cohort of PTC.

*PROX1* expression level in thyroid cancer, as revealed by the above analysis, correlates with the clinical progression. In summary, we observed that higher expression of *PROX1* correlates with a lower stage of tumors which is opposite to the *FGF2* expression level. Moreover, the patients with higher *PROX1* expression have a shorter survival time. These observations can suggest that *PROX1* expression is related to thyroid tumor stage, which was previously shown, e.g., in colorectal cancer and glioblastoma [20,61]. Additionally, we can also suggest that the PROX1:FGF2 expression ratio can play an important role in medical prediction and patient outcomes.

Surprisingly, after addition of FGF2 to CGTH-W-1 cell culture, PROX1 level was significantly enhanced, but still, several selected genes changed in the opposite direction than under PROX1-knockdown. We can assume that PROX1 functions as a transcription repressor for the FGF2 gene, but by itself is upregulated by FGF2 stimulation, thereby contributing to the negative feedback loop of FGF2 signaling in thyroid cancer cells. This possibility is supported by the downregulation of a number of PROX1-regulated angiogenesis-associated genes following the treatment of CGTH-W-1 cells with FGF2. Nevertheless, in a previous study with non-tumoral rat lens explants [31], PROX1 was shown to participate in the positive feedback loop of FGF2 signaling. The reason for this discrepancy is not clear but may be due to the different embryonic origins of cells or their characteristics (normal vs. malignant). 

Since the effect of pro-angiogenic factors is counterbalanced by substances inhibiting angiogenesis, the influence of FGF2 on PROX1 suggests its role in the signaling pathway(s) regulating PROX1 expression, whereas other angiogenesis modulators might be regulated via other signaling pathways. Based on these observations, we can state that PROX1 and FGF2 remain in close relation also in thyroid cancer and regulate two signaling pathways, which cross each other. It seems that this communication between FGF2 and PROX1 may be a very important mechanism regulating the spreading of cancer cells, which can metastasize via the lymphatic or vascular route depending on the analyzed case. 

## 5. Conclusions

In summary, our observations demonstrated that the silencing of PROX1 expression strongly activates angiogenic markers in thyroid cancer cells, which reveals that PROX1 is involved in both processes: lymph- and angiogenesis. We also observed that PROX1 might be in close relation with the pro-angiogenic factor FGF2, which suggests that PROX1 is engaged in the axis of cancer angiogenesis mechanism. Our findings indicate an association between pro-angiogenic factors and distant metastasis of thyroid cancer cells, which can be regulated by changes in PROX1 expression level.

In conclusion, the present study demonstrates that PROX1 is engaged in dissemination of aggressive thyroid cancer types involving the angiogenesis pathway.

## Figures and Tables

**Figure 1 ijms-20-05619-f001:**
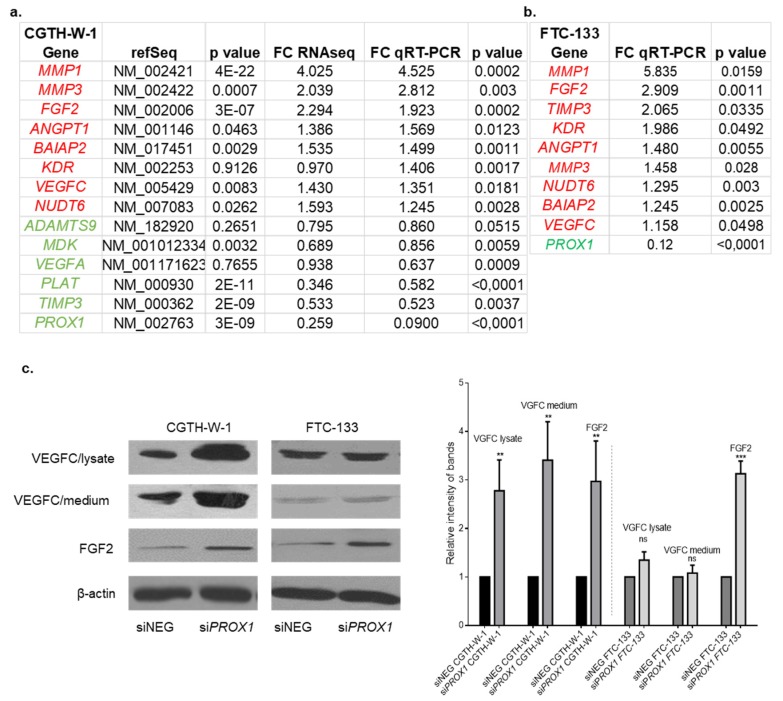
The knockdown of Prospero homeobox 1 gene *PROX1* in CGTH-W-1 and FTC-133 resulted in expression changes of factors involved in angiogenesis. (**a**) The genes involved in the regulation of angiogenesis are significantly regulated under *PROX1*-knockdown in the CGTH-W-1 cell line. The tables show the ID gene (RefSeq), fold changes and *p* values detected in RNAseq analysis, as well as by RT-qPCR technique (the red color indicates up-regulated genes and the green color indicates down-regulated genes). (**b**) Expression levels of selected genes were estimated in FTC-133 cells after silencing of *PROX1* using RT-qPCR. All presented RT-qPCR data represent average of the values obtained from silencing with two si*PROX1* (SA and SC). (**c**) Western blotting was performed on cell lysate and medium collected after 72 h from CGTH-W-1 and FTC-133 cells treated with si*PROX1* or control universal negative siRNA (siNEG). Representative images show the results of Western blotting analysis using the anti-VEGFC (vascular endothelial growth factor C) and anti-FGF2 (fibroblast growth factor 2) primary antibodies; β-actin served as an internal loading control. VEGFC and FGF2 in cell lysates and VEGFC in medium were ~3–4× (** *p* < 0.01) higher after silencing in comparison to control. In FTC-133 cells *PROX1*-knockdown had a negligible effect on VEGFC levels, whereas FGF-2 was ~3.5× (*** *p* < 0.001) increased in FTC-133-si*PROX1*. Signals of Western blotting (WB) were measured using ImageJ and the relative band intensities are shown on the graph. The presented WB intensity data are averages of signals obtained with two si*PROX1* (SA and SC).

**Figure 2 ijms-20-05619-f002:**
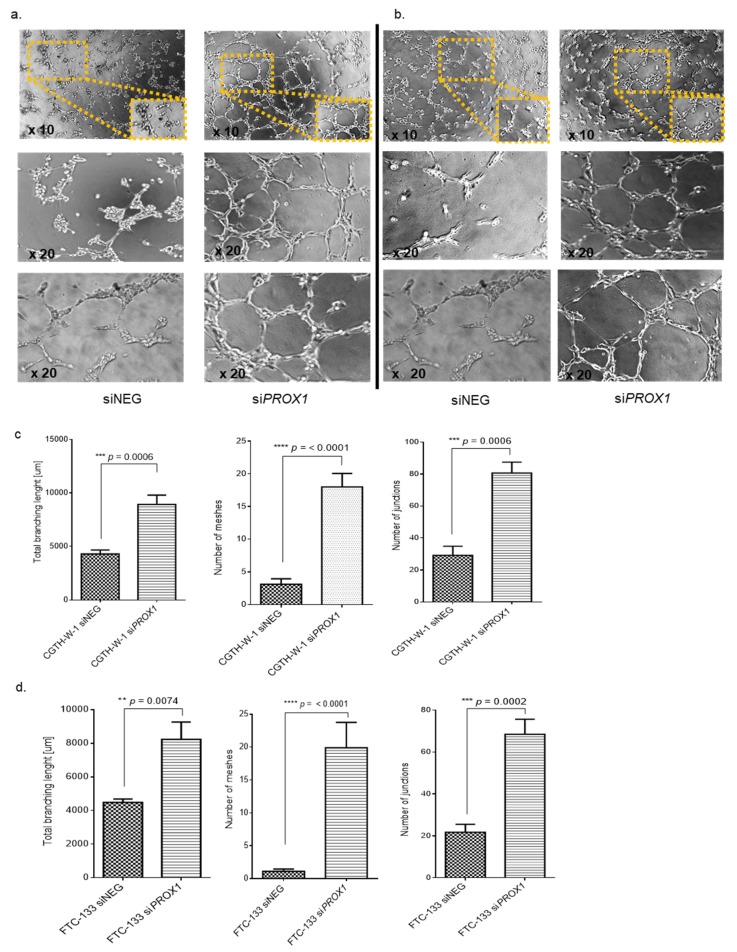
Matrigel tube formation assay. PROX1 silencing in CGTH-W-1 and FTC-133 cells enhances human umbilical vein endothelial cell (HUVEC) angiogenesis in vitro. HUVECs were cultured in 96-well plates coated with a semi-solid Matrigel. The cells were cultured in medium collected from (**a**) CGTH-W-1 and (**b**) FTC-133 cells after silencing of PROX1 and control cells treated with siNEG. The ability of HUVEC cells to form capillary-like structures on Matrigel was assessed under a light microscope after 5 h incubation; original magnifications: ×10 and ×20 lenses. The representative pictures were taken from silencing with si*PROX1* (SA). (**c**,**d**) Total branching, meshes, and junctions were quantified using Angiogenesis Analyzer (ImageJ) and the values were significantly different between HUVEC cultured in siPROX1-medium compared to cells cultured in siNEG-medium collected from both CGTH-W-1 and FTC-133 cells. The presented graphs show the average values obtained from the experiments performed using two si*PROX1* (SA and SC).

**Figure 3 ijms-20-05619-f003:**
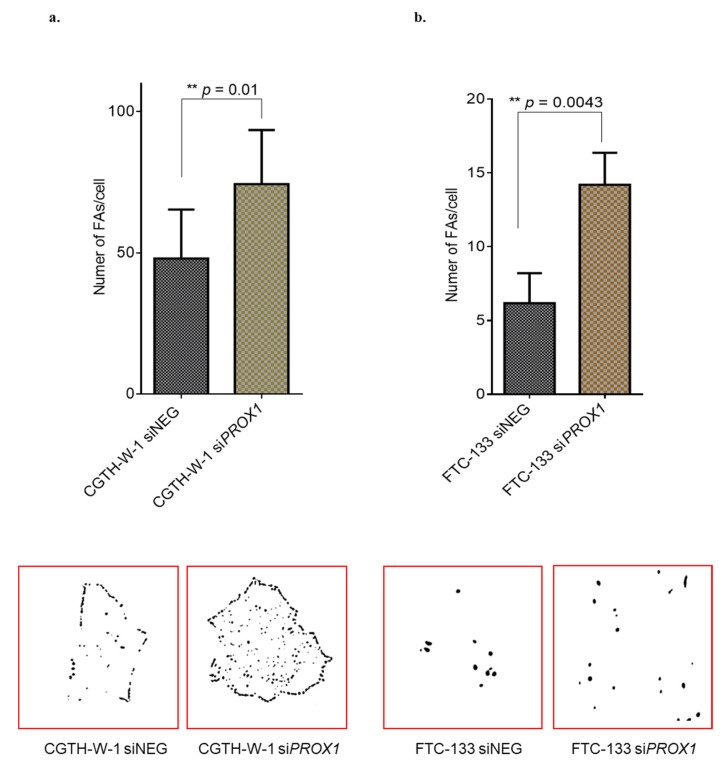
The number of focal adhesions (FAs) increases after silencing of *PROX1* in CGTH-W-1 and FTC-133 cells. FAs were measured using Log3D plugin and CLAHE plugin (ImageJ) according to the described protocol [28]. (**a**,**b**) The number of FAs was significantly higher in cells-si*PROX1* compared to cells-siNEG. Representative images of FAs that were obtained using ImageJ are included below the graph.

**Figure 4 ijms-20-05619-f004:**
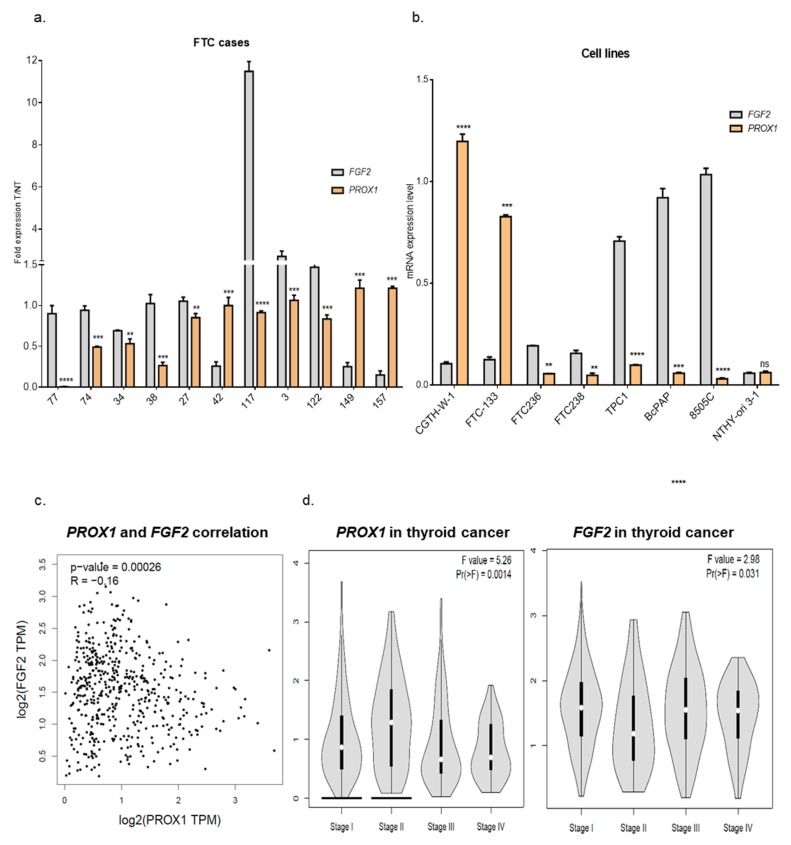
PROX1 expression negatively correlates with FGF2 level in thyroid carcinomas. (**a**) A series of human FTC tissues were analyzed (T/NT) and eight FTC cases show a higher expression of FGF2 compared to PROX1 and three FTC cases (42, 149, and 157) have enhanced expression of PROX1 and reduced level of FGF2. In all tested cases the difference in T/NT ratio between PROX1 and FGF2 was statistically significant (** *p* < 0.01, *** *p* < 0.001, **** *p* < 0.0001) (**b**) PROX1 and FGF2 mRNA expression level in cell lines derived from thyroid carcinomas (SCT derived: CGTH-W-1; FTC-derived: FTC-133, FTC-236, FTC-238; PTC-derived: TPC1, BcPAP; and anaplastic cancer-derived: 8505C) and normal thyroid cells (NTHY-ori 3-1) (**c**) PROX1 expression is negatively correlated with FGF2 in thyroid carcinomas. Data analysis was performed using GEPIA (Gene Expression Profiling Interactive Analysis, http://gepia.cancer-pku.cn/). (**d**) Higher PROX1 expression level is observed in I and II stages of thyroid cancer, and it is downregulated in III and IV stages, whereas FGF2 shows an opposite pattern of expression (GEPIA).

**Figure 5 ijms-20-05619-f005:**
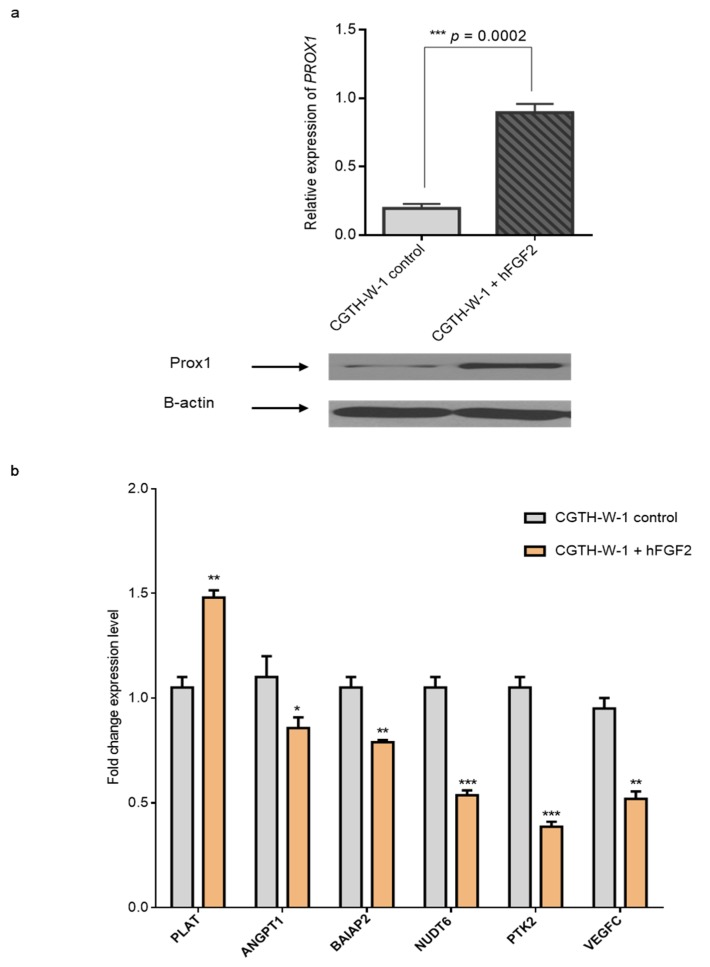
FGF2 induces of PROX1 overexpression in CGTH-W-1 cells. (**a**) Treatment with FGF2 resulted in 4-fold upregulation of *PROX1* mRNA compared to control treated with 0.01% DMSO in RT-qPCR analysis. PROX1 protein overexpression in CGTH-W-1 cells cultured with FGF2 was detected using Western blotting with anti-PROX1 antibody; β-actin served as an internal control. Signals of WB were measured using ImageJ and the final relative quantification values show ~4× higher intensity of PROX1 band after FGF2 treatment in comparison to control. (**b**) Gene expression after FGF2 treatment analyzed by RT-qPCR.

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
