# Peer review of "Transcription Factor Prospero Homeobox 1 (PROX1) as a Potential Angiogenic Regulator of Follicular Thyroid Cancer Dissemination"

_ijms, 2019, doi:10.3390/ijms20225619_

Round 1

Reviewer 1 Report

The manuscript by Ruszinska et al. investigates the relevance of the transcription factor PROX1 in tumor angiogenesis, specifically in follicular thyroid cancers (FTCs).

The impact of PROX1 on tumorigenesis has been debatable and could vary per cancer type.

The authors have previously shown that PROX1 was upregulated in FTC compared to PTC derived cell lines. This proposes the question whether PROX1 might contribute to FTCs having a greater capacity to metastasize to remote tissues via the bloodstream compared to PTCs that migrate through the lymphatic system.

The manuscript examines an important question within the field since it is not well understood what the role of PROX1 is in different tumor types and metastasis. I suggest that the manuscript could be acceptable for publication if comments below are addressed.

It would be informative to show where in the thyroid specimens (Fig 4) PROX1, FGF2, VEGFC, BAIAP2 and PLAT protein are expressed. I understand that tissue availability could be an issue therefore RNA expression has been chosen as the alternative. However, since the tissues that were isolated from thyroid specimens would also contain blood vessels this would impact on RNA expression analysis.  is expressed. Is it upregulated in the tumor cells, in the vasculature or in both? Since the tissues that were isolated from thyroid specimens would also contain blood vessels this would impact on RNA expression analysis.Also, has the RNA sequencing data been normalized against vascular housekeeping genes to correct for the amount of vasculature present in the samples? CGTH cells appear to have more Focal adhesions in the control situation (Fig 3). Does this reflect the behavior of the tumors that these cell lines originate from, i.e rate of metastasis? The authors implicate with this figure the relevance of Focal adhesions upon PROX1 knockdown. Are there also be more Focal adhesions in the tube formation assay? Would a combination of PROX1 siRNA conditioned medium with Integrin signaling inhibitors result in a loss of tube formation. There is an intriguing relationship between PROX1 and clinical progression. It is unclear to me whether with this staging of progression the number of metastatic sites is incorporated since this would strengthen the data. Can the authors clarify this?

Author Response

Subject: “Transcription factor Prospero homeobox 1 (PROX1) as a potential
angiogenic regulator of follicular thyroid cancer dissemination”

Manuscript ID:  ijms-624056

Dear Editor and Reviewers,
Thank you for your very helpful suggestions. We have fully addressed your concerns below and have thoroughly revised the manuscript based on your recommendations.

The manuscript by Rudzinska et al. investigates the relevance of the transcription factor PROX1 in tumor angiogenesis, specifically in follicular thyroid cancers (FTCs).The impact of PROX1 on tumorigenesis has been debatable and could vary per cancer type.

The authors have previously shown that PROX1 was upregulated in FTC compared to PTC derived cell lines. This proposes the question whether PROX1 might contribute to FTCs having a greater capacity to metastasize to remote tissues via the bloodstream compared to PTCs that migrate through the lymphatic system.

The manuscript examines an important question within the field since it is not well understood what the role of PROX1 is in different tumor types and metastasis. I suggest that the manuscript could be acceptable for publication if comments below are addressed.

Dear Reviewer, thank you for your comments, below we split the answers into points, and we did our best to answer your questions.

It would be informative to show where in the thyroid specimens (Fig 4) PROX1, FGF2, VEGFC, BAIAP2 and PLAT protein are expressed. I understand that tissue availability could be an issue therefore RNA expression has been chosen as the alternative. However, since the tissues that were isolated from thyroid specimens would also contain blood vessels this would impact on RNA expression analysis. is expressed. Is it upregulated in the tumor cells, in the vasculature or in both?

Author response: Many genes that canonically regulate lymphatic vessel formation are expressed also in cancer blood vessels and take part in cancer angiogenesis (Stacker SA., et al, 2002). Moreover, many vascular-specific genes, show higher expression also in tumor tissues and regulate the different behavior of cancer cells, such as migration, invasion, adhesion, the proliferation of cancer cells and promote active cancer vascularization.

It is known that in many tumors, the lymphatic and blood vessels are observed in immunohistochemistry staining, but very often with atypical morphology (Senchukova M.A., et al, 2015).

Interestingly, the transcriptome analysis performed using normal liver and malignant hepatoma tissues provided the several genes that were identified as selectively overexpressed in blood vessels during tumor angiogenesis (Bixel MG et al, 2008 ).

However the endothelial cells derived from normal tissue and thyroid cancers were not analyzed in this context, so we are not able to give clear information what is the difference in PROX1, BAIAP2, FGF2 and VEGFC in normal and pathologic vessels. But in our opinion, it is possible that the molecular pattern of cancer vascular system is different than healthy vascularization of the thyroid.

We have previously analyzed PROX1 expression in the thyroid tissue specimens by IHC (Rudzińska, M.et al. IJMS 2019, 20, 2212) and observed that staining was localized in the tumor tissue or the normal tissue in the areas distinct from positively stained lymphoendothelial cells. However availability of the thyroid tissue specimens is limited, what excludes the possibility to perform IHC analysis on the other factors.

Since the tissues that were isolated from thyroid specimens would also contain blood vessels this would impact on RNA expression analysis. Also, has the RNA sequencing data been normalized against vascular housekeeping genes to correct for the amount of vasculature present in the samples?

Author response: RNA sequencing was performed on cell lines, which helped to avoid this problem. We analyzed 2 different FTC cell lines derived from different metastatic sides, so the observed effect seems to be very specific for this type of cancer cell and not disrupted by the vascular component.

CGTH cells appear to have more Focal adhesions in the control situation (Fig 3). Does this reflect the behavior of the tumors that these cell lines originate from, i.e. rate of metastasis?

Author response: Yes, CGTH cells are considered to be more aggressive cells because they come from sternum metastasis, whereas FTC-133 cells are derived from lymph node metastasis. Moreover, on the basis of our data (compare Rudzińska, M.et al. IJMS 2019, 20, 2212 for CGTH-W-1 cells and Rudzińska, M.et al. Oncotarget 2017, 8, 114136 for FTC-133 cells) we can conclude that indeed CGTH-W-1 cells might have higher metastatic potential as they show higher invasion potential than FTC-133 cells.

The authors implicate with this figure the relevance of Focal adhesions upon PROX1 knockdown. Are there also be more Focal adhesions in the tube formation assay?

Author response: The tube formation assay is performed on HUVEC cells in the 3D culture  in semi-solid Matrigel, so it is technically impossible to measure focal adhesions in the format we utilize in our laboratory. Specifically, in the results reported in the manuscript we analyzed the active form of focal adhesion kinase in adherent cells, attached to the coverslips, using immunofluorescence staining. On the other hand, the angiogenesis in vitro software allows measuring the number of branching (the nodes connected to tree different line segments, which indicates the new vessels sprout), meshes/loops, and junctions between the endothelial cells created in the Matrigel matrix, not on fixed and stained coverslips.

Would a combination of PROX1 siRNA conditioned medium with Integrin signaling inhibitors result in a loss of tube formation.

Indeed, in the previous study we observed increased levels of a number of molecules regulating the focal adhesion, including integrins. Adhesive molecules like caveolins, collagens, and integrins are actively involved in focal adhesion and thus angiogenesis (Lechertier T et al., 2012).  For example, the αvβ3 Integrin Antagonist S247 decreases the angiogenesis in vitro and in vivo (Niels Reinmuth et al., 2003). On the other hand, studies using β3- and β5-integrins null mice demonstrated enhanced tumor growth, tumor angiogenesis, where vascular endothelial growth factor (VEGF)-A–induced vascular permeability caused by elevated levels of VEGF receptor (VEGFR)-2 on ECs. The authors explained that it could be a compensatory effect (Ganapati H. Mahabeleshwar et al., 2006).

All above data suggested that angiogenesis is a very complex mechanism, and many factors take part in this process, so it is a reason why cancer angiogenesis is still unclear, and new investigation can give valuable information in this field.

There is an intriguing relationship between PROX1 and clinical progression. It is unclear to me whether with this staging of progression the number of metastatic sites is incorporated since this would strengthen the data. Can the authors clarify this?

Author response:

As we understand reviewer’s question relates to the analysis presented on Figure 4d and Figure S4. These analyses involved publicly available data analysis tools GEPIA (http://gepia.cancer-pku.cn/) and The Human Protein Atlas (https://www.proteinatlas.org/ENSG00000117707-PROX1/pathology/thyroid+cancer), respectively that both use data available from The Cancer Genome Atlas (TCGA) database. We believe that staging in the above dataset is based on the guidelines of the American Joint Committee on Cancer/Tumor-Node-Metastasis (AJCC/TNM) staging system.

Reviewer 2 Report

The authors demonstrated the role of PROX1 in thyroid cancer. PROX1 is novel and interesting, however, some point should be revised as following:
Introduction: The goal of this study should be emphasized. And, possible results or hypothesis (the impact of PROX1 in thyroid cancer) should be presented.

Reference: Too many reference! In Introduction, there are 35 references. Totally, Introduction and Discussion are lengthy.

Based on these concerns, I recommend the publication of this work with Minor revision.

Author Response

Subject: “Transcription factor Prospero homeobox 1 (PROX1) as a potential
angiogenic regulator of follicular thyroid cancer dissemination”

Manuscript ID:  ijms-624056

Dear Editor and Reviewers,
Thank you for your very helpful suggestions. We have fully addressed your concerns below and have thoroughly revised the manuscript based on your recommendations.

The authors demonstrated the role of PROX1 in thyroid cancer. PROX1 is novel and interesting, however, some point should be revised as following:

Introduction: The goal of this study should be emphasized. And, possible results or hypothesis (the impact of PROX1 in thyroid cancer) should be presented.

Author response: Thank you for this comment which helped to improve this part of the manuscript. The added part is marked in red.

Reference: Too many reference! In Introduction, there are 35 references. Totally, Introduction and Discussion are lengthy.

Author response: Thank you for your comment, which helps us to avoid the redundant parts in our paper. The one part in the introduction has been already deleted. The number of references in the introduction and discussion is reduced. The reason why discussion can look quite long is that we would like to introduce all genes involved in the detected changes (connected with angiogenesis), what in our opinion makes it easier for readers. However, we did our best to make the introduction and discussion shorter with fewer numbers or citation